# Development and Characterization of a Self-Tightening Tourniquet System

**DOI:** 10.3390/s22031122

**Published:** 2022-02-01

**Authors:** Saul J. Vega, Sofia I. Hernandez-Torres, David Berard, Emily N. Boice, Eric J. Snider

**Affiliations:** U.S. Army Institute of Surgical Research, JBSA Fort Sam Houston, San Antonio, TX 78234, USA; saul.j.vega.ctr@mail.mil (S.J.V.); sofia.i.hernandeztorres.ctr@mail.mil (S.I.H.-T.); david.m.berard3.ctr@mail.mil (D.B.); emily.n.boice.ctr@mail.mil (E.N.B.)

**Keywords:** hemorrhage, tourniquet, closed-loop, automation

## Abstract

Uncontrolled hemorrhage remains a leading cause of death in both emergency and military medicine. Tourniquets are essential to stopping hemorrhage in these scenarios, but they suffer from subjective, inconsistent application. Here, we demonstrate how tourniquet application can be automated using sensors and computer algorithms. The auto-tourniquet self-tightens until blood pressure oscillations are no longer registered by the pressure sensor connected to the pneumatic pressure cuff. The auto-tourniquet’s performance in stopping the bleed was comparable to manual tourniquet application, but the time required to fully occlude the bleed was longer. Application of the tourniquet was significantly smoother, and less variable, for the automatic tourniquet compared to manual tourniquet application. This proof-of-concept study highlights how automated tourniquets can be integrated with sensors to provide a much more consistent application and use compared to manual application, even in controlled, low stress testing conditions. Future work will investigate different sensors and tourniquets to improve the application time and repeatability.

## 1. Introduction

Uncontrolled hemorrhage remains a major challenge for both civilian and military medicine. In fact, it is currently the leading cause of death for combat casualty care [1] and the second highest cause of traumatic death in the civilian sector [2]. Further, penetrating trauma due to shrapnel, bullets, or other projectiles are often lethal in rural, remote, or military settings where access to appropriate medical resources is limited [3,4,5]. In the case of injured extremities, tourniquets (TKTs) are used to stop or at least slow the rate of blood loss distal to the tourniquet by applying mechanical pressure to occlude arterial blood flow.

While TKTs are critical, life-saving medical devices, they can be challenging to apply properly [6,7]. For extremity TKTs, cases of incorrect application or insufficient tightening occur, thus failing to control bleeding in a timely fashion, particularly among untrained users [7]. Even among trained personnel, TKT application can be variable depending on experience and skill level [6]. Moreover, TKT overtightening can result in damage to underlying nerves, muscles, blood vessels, and soft tissues which can contribute to patient morbidity [8]. In any case, for these devices to be effective, they must be applied properly and within minutes of the initial injury, prior to the onset of hemorrhagic shock [9]. These challenges with TKT use become further magnified in high stress situations where more than one hemorrhage casualty may be present, which is especially true for military combat casualty care [10]. Improved TKT designs are needed that can perform reliably and more consistently to assist medics in the high stress situations where these devices are employed.

One approach to improve TKT application inconsistencies is to automate their operation through the integration of sensors and computer algorithms. Autonomous medical systems in the battlefield have the potential to ease the management of complex clinical scenarios by providing continuous assessment of patient status, clinical guidance and decision support, and automatically performing life-saving actions in order to provide precious additional minutes to the combat medic or lifesaver to intervene [11,12]. These systems will reduce cognitive and physical load for these caregivers, allowing for improved outcomes, particularly in mass-casualty situations where medics may find themselves overwhelmed and with limited resources.

Commercially available surgical tourniquets automatically activate to occlude blood flow to a limb by relying on detection of blood flow in the distal portion of the extremity (e.g., a finger). Such approaches may not work for a sufficiently large bleed that precludes perfusion to the tissue monitored by the sensor or when the extremity in question has suffered a traumatic amputation. More recently, so-called “personalized tourniquets” have been marketed for applications in physical therapy and rehabilitation. These devices do not require monitoring of distal blood flow but instead sense flow directly at the point of tourniquet application, often using an oscillometric approach [13]. Here, we detail the design and characterization of a self-tightening, automated pneumatic TKT (aTKT) system using the same oscillometric principle to manage uncontrolled extremity hemorrhage.

## 2. Materials and Methods

### 2.1. Overview of the Self-Tightening aTKT System

For this work, a proof-of-concept, self-tightening tourniquet system (aTKT) was designed with the goal of compressing a bleeding extremity until the system automatically detects that blood flow to the limb has stopped, and thus the hemorrhage has stopped as well. Specifically, our device consists of a computer-controlled pneumatic pump that inflates a cuff wrapped around an injured limb and continues to inflate this cuff to increasing levels of air pressure in order to provide compression to that extremity. The computer analyzes a signal from the tourniquet to determine whether blood is flowing to the limb and continues to activate the air pump to further compress the extremity until blood is no longer flowing into it; at this point, the computer deactivates the pump. The system also includes an air release valve to allow the computer to completely disengage the tourniquet, if so required.

### 2.2. Oscillation Detection Algorithm

The detection of blood flow in our design employs an oscillometric approach similar to that used in automated blood pressure monitors: an air pressure transducer attached to the pneumatic cuff (length = 45.72 cm [18 in, arm] or 86.36 cm [34 in, leg], Stryker, Kalamazoo, MI, USA) detects vibrations transmitted to the cuff through the subject’s skin due to the blood flowing in the underlying arteries. In our system, a computer algorithm was developed to analyze this signal and determine whether blood was flowing to the limb of interest. The algorithm determines that blood is no longer flowing when it ceases to detect vibrations in the pneumatic cuff (Figure 1A).

The aTKT algorithm was written in Python 3.8 and employs routines from the open-source Python library for scientific computing, SciPy. It runs on a standard personal computer that interfaces with the air pressure transducer (MPX5050DP, NXP Semiconductors, Eindhoven, The Netherlands) through a data-acquisition unit (LabJack Corporation, Lakewood, CO, USA). The same LabJack device also allows the algorithm to activate and deactivate both the air pump (Dewin, Amazon, Seattle, WA, USA) and solenoid valve (EndlessParts, Amazon, Seattle, WA, USA). The selection of the pressure sensors was made primarily on the basis of its pressure operating range (0–375 mmHg) to ensure it covered the expected range of operation of the tourniquet, allowing for maximum signal resolution. In addition, the pressure sensor had simple circuitry requirements for interfacing with the data-acquisition unit. Calibration was performed at the beginning of each activation of the device, when the system zeroed the transducer reading at atmospheric pressure while the pneumatic cuff was still fully deflated.

When active, the aTKT control algorithm samples the signal coming from the transducer at 50 Hz and then starts a stepwise process of activating the air pump to increase the air pressure in the pneumatic cuff in increments of 15 to 30 mmHg. Between each stage, the algorithm pauses inflation and collects a 3-s-long sample of pressure readings. This sample is processed for noise removal through a 5th-order Butterworth digital filter with band-pass frequencies between 1 Hz and 5 Hz, and then is further processed by a peak detector function to identify possible oscillations. This function relies on peak properties to identify peaks in the input stream, in particular their “prominence” and distance between peaks. The specific values for these properties were empirically determined for each of our leg and arm models. The algorithm then compares the number of peaks found in the reading sample to an experimentally determined threshold to determine whether blood flow is being registered by the pneumatic cuff. Should the count of peaks exceed that threshold, the algorithm then re-activates the pneumatic pump to continue inflating the cuff to the next pressure step.

This process of inflating the cuff and sampling the air pressure repeats itself until the algorithm ceases to detect oscillations in the sampled signal. Using additional criteria (e.g., how many consecutive times were oscillations not found, etc.), the algorithm makes a determination on whether blood flow in the extremity is completely occluded, at which point the system ceases inflating the pneumatic cuff and provides an audible and visual notification indicating that the tourniquet is fully engaged.

### 2.3. Perfusion Model for Tourniquet Evaluation

For testing our tourniquet device, we utilized a synthetic human arm and leg model developed for physiological relevant testing (SynDaver Inc., Tampa, FL, USA). The arm or the leg were attached to a modified water pump to simulate cardiovascular function from 80 to 200 BPM for this study. The SynDaver synthetic model has anatomically relevant features for use in this study such as a physiologically relevant vasculature with brachial or popliteal arteries, hemorrhage sites in the arm or leg, and a region for applying a TKT to stop hemorrhage (Figure 2). A patient monitor (Infinity Delta XL Dräger, Lübeck, Germany) was used to assess blood pressure in the system through two transducers (ICU Medical, San Clemente, CA, USA) connected proximal and distal to the tourniquet. All data was recorded by using a data acquisition system (PowerLab, ADInstruments, Sydney, Australia) attached to the patient monitor and other sensor inputs. Unfortunately, the mean arterial pressure and overall waveform were significantly outside the desired physiological range of values using the modified water pump as intended (MAP > 200 mmHg, data not shown). As a result, the SynDaver arm and leg were fitted with a bypass loop utilizing a gate valve to allow a fraction of the flow to bypass the arm and leg, reducing the overall MAP of the system (Figure 2C). A resizable air canister was incorporated to provide vibration damping and reduce the amplitude of the pressure waveform. Controlling the bypass and air canister volume allowed to obtain physiologically relevant pressures in the arm and leg. A target MAP of 95 mmHg was set, respectively.

Bleed points in the synthetic arm and leg systems consisted of a 3-way valve that could be opened to divert a fraction of the flow out of the system to mimic a bleed. To assess multiple bleed rates, 3 different blunt needle sizes (20, 17, and 14 gauge, McMaster-Carr, Elmhurst, IL, USA) were attached to alter the fluidic resistance and, thus, the fraction of fluid leaving the system. Bleed rates were quantified at 4 heart rates (80, 120, 160, and 200 BPM) by collecting the effluent for 30 s and recording the mass. These values were compared to inlet flow rates to the arm or leg, as determined by a flow sensor (SynDaver Inc., Tampa, FL, USA).

### 2.4. Tourniquet Evaluation Experimental Design

Performance of the auto-tourniquet (aTKT) system was compared to a manual tourniquet (mTKT) application in both the arm and leg synthetic systems. An expert in TKT application was utilized for mTKT application using a Combat Application Tourniquet (CAT; North America Rescue, Greer, SC, USA). The arm and leg systems were evaluated at each of the aforementioned heart rates with the systems tuned to be within target blood pressure ranges downstream from where the tourniquet was applied. Each bleed rate was initiated and effluent from the bleed site was collected undisturbed for 30 s. The aTKT or mTKT was then applied as described in their respective sections below, and once application was complete, the system was allowed to stabilize for 30 s, at which time a second collection of any remaining bleed effluent was performed for another 30 s. Successful aTKT or mTKT application was defined as 95% bleed reduction after tourniquet application. As it was not possible to distinguish aTKT or mTKT application error being due to the tourniquet alone or the model, only successful applications were evaluated further. For some experiments, bleed effluent was collected for 5 min to determine if the bleed rate increased with time. Due to limitations of the SynDaver system and its risk of drying out, application beyond 5 min was not viable. In addition to bleed effluent collection, blood pressure data was collected downstream of the tourniquet. Further, TKT applied stress data was recorded by placing a force sensor (Tekscan, Boston, MA, USA) under the aTKT and mTKT during application.

To evaluate TKT overtightening, a “theoretical” bleed stop was determined by evaluating the relationship with respect to time and MAP during TKT application. Theoretical bleed stop time was defined as when the second derivative of MAP with respect to time approaches zero after beginning TKT application, indicating that TKT compressions are no longer impacting MAP. The applied TKT stress at this time was compared to the maximum TKT stress to determine how much excess mechanical stress (i.e., overtightening) was induced by the mTKT or aTKT. 

#### 2.4.1. Auto-Tourniquet Application

During the test, the pneumatic cuff was pre-placed high above the simulated bleeding site, and then bleeding would be initiated. Moments later, the tourniquet algorithm was manually started, and the computer was allowed to take over control of the device. At that point, through the tourniquet, the computer increased pressure to progressively compress the limb until the algorithm determined that the hemorrhage had been stopped.

#### 2.4.2. Manual-Tourniquet Application

Application was performed similarly with a mTKT for a point of comparison for the aTKT. A combat application tourniquet (CAT) was used as it is widely used in the pre-hospital setting [14,15]. This mTKT has a strap width of 1.5 inches (3.8 cm) and uses a one-handed, windlass system to occlude blood flow. For this project, the mTKT was placed at a similar location on the arm or leg SynDaver test system to the aTKT and tightened. Once hemorrhage was initiated, the windlass was rotated until hemorrhage was reduced to approximately one drop per second. Afterwards, the windlass was locked into place using the clip on the mTKT.

### 2.5. Statistical Analysis

At least triplicate runs were performed at each heart and bleed rate for both the aTKT and mTKT. Results are shown as mean and standard deviation throughout. Coefficient of Variation (CV), defined as the standard deviation relative to the mean, was calculated for test metrics to better present the application variability for both the aTKT and mTKT. When calculating CV, all arm or leg data were used across all heart and bleed rates to generate larger data sets and evaluate the variability across all testing parameters. Statistical significance was determined (GraphPad Prism 9.1.0) by two-way ANOVA post hoc Sidak multiple comparisons test to compare performance between mTKT and aTKT at each of three bleed rates for both the arm and leg, where *p* < 0.05 indicated significant results. For determining statistical significance for differences between mTKT and aTKT CV values, student’s unpaired *t*-test was used to evaluate significance, where *p* < 0.05 indicated significance. *p*-values less than 0.05 (*) were used to indicate significance and are indicated throughout.

## 3. Results

### 3.1. Overview of Self-Tightening Tourniquet System

The proof-of-concept, self-tightening tourniquet used in our study consisted of a computer-controlled pneumatic pump that inflated a cuff wrapped around an injured limb, applying increasing levels of compression to that extremity (Figure 1A). To automate the tourniquet tightening process, this design employed an oscillometric approach to detect blood flow similar to that used in automated blood pressure measuring devices [16]. To do this, an algorithm analyzed the air pressure in the pneumatic cuff to determine whether blood was flowing to the limb. Concisely, an air pressure transducer measured the vibrations in the pneumatic cuff that are generated by the pulsatile blood flow in the subject’s underlying arteries and were transmitted to the cuff (Figure 1B–D). During its operation, the aTKT activated the air pump to inflate the pneumatic cuff (Figure 1B) tightening around the extremity until the vibrations were stopped. This results in reduced mean arterial pressure (MAP) distal of the TKT site (Figure 1C) and hemorrhage rate reduction greater than 95% (Figure 1D), at which point the air pump was deactivated. The system also included an air release valve to allow the computer to completely disengage the tourniquet if so required.

### 3.2. Syndaver Tourniquet Test Platform

In order to test the performance of the aTKT, we used a synthetic model of a human arm and leg (SynDaver, Tampa, FL, USA) (Figure 2). These synthetic systems have physiologically relevant vasculature and anatomical features that allow evaluation of the aTKT system’s ability to occlude blood flow. To mimic cardiovascular physiology, a modified water pump was used to perfuse water into the system creating pulsatile flow, resembling a human arterial waveform in shape and magnitude (Figure 3A,B). A bypass loop was constructed to allow for circulating flow after TKT application (Figure 2C). Each synthetic model has physiologically relevant anatomy for TKT application, a single downstream “bleeding” site manually operated through a 3-way valve to simulate hemorrhage, and each was fitted with a pressure transducer (PT). Three different controlled hemorrhage rates were introduced in the system (by separately attaching 20G, 17G, and 14G blunt needles to the bleeding valves) to create various scenarios for comparing the performance of the aTKT to a manually applied Combat Application Tourniquet (mTKT) (CAT, North America Rescue, Greer, SC, USA) (Figure 3C,D).

### 3.3. Performance of Self-Tightening Tourniquet Sytem

#### 3.3.1. Stop the Bleed Performance

First, overall “stop the bleed” capabilities were evaluated for both the aTKT and mTKT at three hemorrhage rates (Figure 3C,D) and 4 heart rates (80, 120, 160 and 200 bpm). A successful TKT application was established to be that in which the hemorrhage rate was reduced by at least 95% from the initial rate 30 s after tourniquet tightening. As with real world scenarios of TKT application, aTKT and mTKT application in these platforms were variable and sometimes resulted in failure to stop the bleed—only data from successful applications was evaluated. Although the aTKT system was capable of meeting the 95% bleed reduction threshold for both the arm and leg test platforms and for each hemorrhage rate (Figure 4A,B), its performance on the arm platform resulted in a significantly higher hemorrhage post-application compared to the mTKT for the 17G (97.7% aTKT vs. 99.5% mTKT hemorrhage reduction) and 14G (96.9% aTKT vs. 99.2% mTKT hemorrhage reduction) bleed rates. However, a significant difference in performance between the two tourniquets was not observed when testing with the leg model. In addition, we evaluated a hemorrhage rate 5 min post TKT application for both the mTKT and aTKT and found that the aTKT system resulted in significantly higher hemorrhage rates for both extremities, even failing to meet the 95% hemorrhage reduction threshold (Figure 4C), indicating the aTKT was not as effective as the mTKT in keeping the flow occluded over longer durations. In summary, the aTKT system stopped the bleeding for both the arm and leg test systems but was less effective than the mTKT in some scenarios.

#### 3.3.2. Tourniquet Application Stress

We next compared the mechanical stress applied to the arm or leg under the TKTs using force sensors. There was significantly less stress applied by the aTKT compared to the mTKT for both the arm and leg (Figure 5A,B). However, the TKT width for the aTKT was approximately 10 cm (4 inches) while the mTKT was 3.8 cm (1.5 in) in width, and given the known effect TKT width has on application stress, it is difficult to evaluate these differences [17]. Instead, the consistency in applied stress was evaluated as automating the manual processes should greatly improve reproducibility. Variability of stress was quantified through calculating a coefficient of variation (standard deviation of the stress relative to the mean) for both the arm and leg for each hemorrhage rate (Figure 5C). There was a significant reduction in variability when comparing the aTKT to mTKT stress results (16.2% aTKT vs. 43.8% mTKT coefficient of variation), highlighting the effect computer control had on the reproducibility of complex, subjective medical procedures. Further, while applied stress increased for bleed severity for the leg aTKT scenarios, the increase in stress was not as pronounced for the arm aTKT scenarios. The mTKT increased in stress with bleed severity for both the arm and leg. This application stress difference for the arm testing scenario may explain the mTKT and aTKT hemorrhage stop performance differences observed (Figure 4A).

#### 3.3.3. Application Time

Similarly, application time was compared for both TKT types. For the aTKT, application time was determined internally by the closed-loop algorithm analyzing the signal from the air pressure transducer inside the aTKT cuff. For the mTKT, application time was qualitatively set by initial windlass turn to the point when bleeding from the hemorrhage site was observed to have stopped. Application times started after the TKT was strapped around the extremity, and only accounts for TKT tightening as time for TKT placement on the limb was too variable. The mTKT was significantly faster to apply compared to the aTKT for both the arm and leg (Figure 6A,B). However, similar to application stress, variability of application time was significantly reduced for the aTKT compared to the mTKT (3.7% aTKT vs. 20.3% mTKT coefficient of variation, Figure 6C). Overall, while the aTKT had longer application times, its automated operation provided a more consistent, objective process that may be optimal in high casualty, high stress environments.

#### 3.3.4. Tourniquet Overtightening

One potential issue with manual TKT application is overtightening that could unnecessarily damage underlying tissue. To evaluate overtightening by the aTKT, we determined when the changes in MAP were stabilized during TKT application by using the second derivative of MAP vs. time (Figure 7A). The time at which the second derivative of MAP reached zero was set as the “theoretical” bleed stop as further TKT tightening had minimal impact on MAP. An example of how theoretical bleed stop was determined is shown (Figure 7A). For each theoretical bleed stop, the corresponding applied TKT stress was determined and then compared to the maximum stress applied, in order to determine how much excess mechanical stress was applied. Using this method, arm testing showed much higher, significant overtightening by the mTKT compared to the aTKT (Figure 7B). For the leg, the mTKT was significantly more overtightened compared to the aTKT at slow 20G bleed rates, but comparable for both higher bleed rates (Figure 7C). On average, across all test conditions for the arm and the leg, the mTKT was 91% overtightened while the aTKT was only 25% overtightened. In summary, the aTKT outperformed the mTKT in terms of reducing TKT overtightening for nearly all testing scenarios, highlighting a significant advantage for medical device automation.

## 4. Discussion

Tourniquets are life-saving devices in severe trauma instances encountered in emergency medicine as well as combat casualty care in military medicine. However, application of a TKT is not trivial. Overtightening can cause irreversible damage to the occluded or compressed tissue. Conversely, a TKT applied too loosely may not adequately stop the hemorrhage to stabilize the casualty. These errors are amplified during mass casualty, high stress scenarios. To simplify the process, automation could reduce the reliance on human input to apply a TKT using physiological sensors to monitor the application of the device. Towards this, the self-tightening aTKT system highlighted in this work is an important step towards automating this life-saving medical tool. 

The aTKT operation relies on the oscillometric principle, in which fluctuations in air pressure generated by arterial blood flow are measured in the pneumatic TKT. This is a simple, non-invasive physiological input for controlling TKT application, and we were able to show that closed-loop TKT application following this input was able to stop hemorrhage comparable to mTKT application. This is a proof-of-concept study for the utility of intelligent, closed-loop TKT systems. The air pressure sensor input used with the aTKT relies on a pneumatic TKT design. Therefore, for automating other mechanical TKTs, different sensor inputs will need to be utilized for controlling TKT application. 

Regardless, the significance of automating TKT application was shown in this study as its operation was much more consistent, and overtightening was lessened compared to mTKT application. Application consistency was quantified for both the TKT application time, and the mechanical stress applied to the underlying tissue by the TKT. Variability in application time was reduced by 50% for leg application and as much as 80% for arm testing. Similarly, mechanical stress induced by the aTKT was significantly less variable compared to that of the mTKT for both the arm and leg testing setups. In addition, TKT overtightening was significantly reduced for the aTKT compared to the mTKT in nearly all test scenarios. Of note, improvements of this magnitude were in a controlled testing situation and will likely be orders of magnitude increased in the high stress situations in which TKTs are normally employed. Further, this variability was observed in a single experienced person applying a mTKT and, as such, will likely increase when factoring in variability across multiple users, including novices. By automating TKT application, the subjective nature of TKT application is objectified and will be minimally impacted by different user applications. 

While the aTKT system was more consistent in operation, other aspects of its use compared unfavorably to the mTKT. First, the overall application time was considerably slower, more than doubling the application time for the arm and leg test setups. Although the inflation protocol can be tuned to reduce the overall application time, the system is relying on sensor input to stop the bleed but not excessively overtighten to preserve underlying tissue. The feedback mechanisms in the closed-loop algorithm therefore require application to be slower. This is further exacerbated as the algorithm employs a simple approach that does not allow for reliably analyzing oscillometric pressure readings while the cuff is inflating, requiring pauses between each TKT pressure increase. A more sophisticated algorithm, and possibly inclusion of other sensing inputs that are independent of the tightening mechanism may allow this problem to be resolved in future automated TKT designs. Second, there were consistently higher bleed rates after aTKT application compared to mTKT. As the aTKT is trying to avoid overtightening beyond the point of bleed stop, higher bleed rates are to be expected. Nevertheless, the aTKT was still capable of decreasing the bleed by more than 95% immediately after TKT application. Increased tightening can be accomplished by adjusting the algorithm sensitivity to oscillometric measurements.

In addition, bleed rate was more pronounced for the aTKT compared to the mTKT longer durations post-tourniquet application. The aTKT resulted in bleed rates beyond the 95% bleed rate reduction window while the mTKT did not. In its current iteration, the aTKT senses loss of oscillation and immediately ceases additional tightening of the tourniquet. A more complete design of the control algorithm that was explored for demonstration purposes but not systematically tested for the work presented here, included continued monitoring after TKT engagement, which allowed it to re-tighten the cuff if oscillations re-initiated (indicating TKT loosening). This feature could offer a particular advantage to an automated system as TKT devices may loosen due to external mechanical stresses during casualty management and evacuation, and relaxation of the compressed tissue over time. An intentional periodic loosening can also be integrated into the algorithm allowing for reperfusion of viable tissue distal to the TKT site. This can be beneficial for instances where prolonged TKT use is required to prevent tissue necrosis, extensive nerve damage, and potential limb preservation once surgical intervention is possible [18,19,20].

There were some limitations with the current study. First, while the SynDaver allowed for laboratory testing on human anatomy without the need for human tissue, the system was not optimal. Physiological cardiovascular waveforms were challenging to reproduce with the focus instead on trying to keep MAP as physiological as possible while systolic/diastolic features remained overly pronounced. The result was a system that allowed proof-of-concept evaluation only. Next steps will transition this into more physiologically relevant models and eventually animal testing. We have recently developed a silicone model of the arm that allows physiologically relevant pulsatile flow that enables more consistent TKT application [21]. Second, placement of the TKT was limited to a very narrow section of the model that provided meaningful compression of the vasculature; however, this was not always the ideal distance from the hemorrhage site. Future work evaluating how the aTKT response differs based on location applied will need to be evaluated. Third, due to the pump rates being too high for creating physiological MAP, a bypass loop from the synthetic extremity was required (Figure 2). After TKT application, more flow was diverted to the bypass increasing pressure upstream of the TKT. While not physiological, this likely resulted in more pressure held back by the TKT than would be needed which may alter the results of the study when moving into animal studies. Lastly, for this pilot study, only a single type of mTKT and single expert operator were used as a comparison point. Other TKT designs may compare differently to the aTKT system; however, we considered it not prudent to introduce more factors into this initial pilot study. Additional designs and operators will be utilized in follow-up studies.

In the future, different designs for tourniquet devices should be explored, as pneumatic tourniquets have the inherent disadvantage that damage to the cuff could easily render them inoperable. A mechanical system (e.g., winch-based) might be more reliable. Either tourniquet type would benefit from novel sensors and improved sensor placement to reduce signal noise. In addition, improved signal processing would further assist with these challenges in future aTKT designs. The current aTKT design would receive large signal artifacts due to patient motion or during patient transport. Ultrathin, e-tattoos sensors currently being developed by academia might offer a solution [22,23,24]. Future work could also involve fully automating tourniquet deployment by integrating the device to a system capable of identifying a sudden hemorrhage in a limb (e.g., through detection of changes in Doppler phasicity of blood flow), and even integrating with systems for the management of hemorrhagic shock.

## 5. Conclusions

This study provides proof-of-concept for an automated tourniquet design. The system was able to achieve comparable but more consistent levels of performance to that of an expert in the application of an extremity tourniquet. The ability of such a device to immediately, effectively, and consistently arrest uncontrolled bleeding might be particularly beneficial to an injured service member, confronted with the stressful situation of a massively hemorrhaging limb, perhaps with limited mobility and/or under fire. This system could also be useful in mass-casualty situations where an overwhelmed combat medic or civilian caregiver must manage multiple bleeds simultaneously, or even to a novice combat lifesaver who has no real-life experience in the application of a tourniquet in those same stressful scenarios.

## Figures and Tables

**Figure 1 sensors-22-01122-f001:**
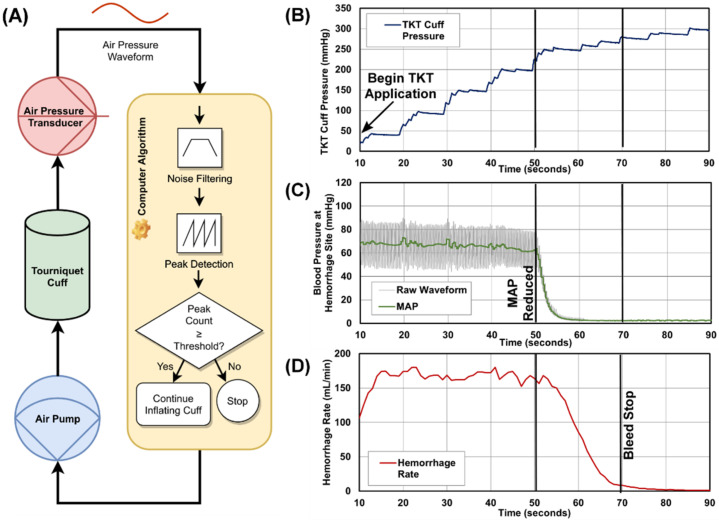
Overview of the Self-Tightening Auto-Tourniquet (aTKT) system. (**A**) Process flow diagram for the aTKT algorithm mechanism. An air pressure sensor attached to the pneumatic pressure cuff sensed pressure oscillations and continued tightening through a closed-loop feedback mechanism until these oscillations were not detected. (**B**–**D**) Representative results for the aTKT for (**B**) air pressure in the pneumatic cuff, (**C**) arterial waveform data, and (**D**) hemorrhage rate vs. time. The aTKT was engaged at time shown on the plot and reduction in MAP and bleed stop are noted on the plots.

**Figure 2 sensors-22-01122-f002:**
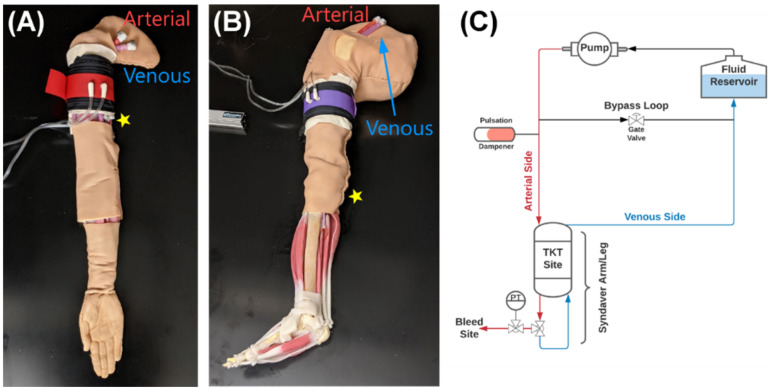
Overview of the human synthetic aTKT test platform. Synthetic (**A**) arm and (**B**) leg structures (SynDaver) served as the basis for testing the aTKT system as they have physiological and anatomical vasculature. Approximate aTKT position for each SynDaver model as well as arterial and venous tubing connections are shown. Approximate position of the bleed site is indicated by star label. (**C**) Flow diagram of the test platform. A reservoir filled with water supplied fluid to a cardiovascular-mimicking pump, which provided flow to either the arm/leg or a bypass loop that prevented over-pressurization when TKT was applied. Hemorrhage site was located at a valve in the arterial side of the arm/leg, and TKT was applied proximally to the hemorrhage site. A pressure transducer (labeled PT) was attached at the hemorrhage site.

**Figure 3 sensors-22-01122-f003:**
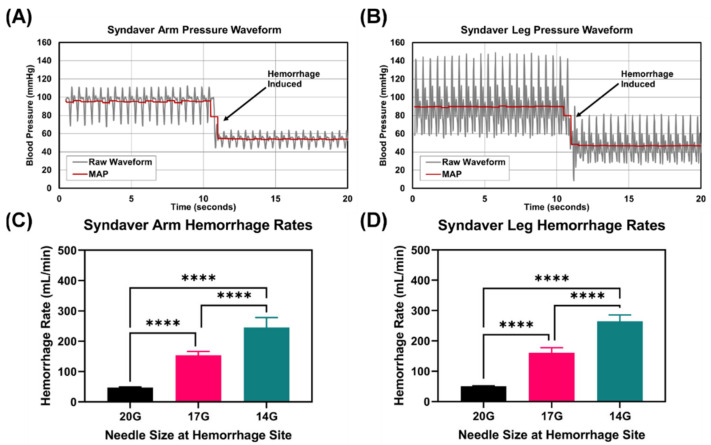
Physiological Features of the TKT test platform. Representative raw pressure waveform data and mean arterial pressure in the synthetic (**A**) arm and (**B**) leg. Hemorrhage induction is labeled and the effect on the pressure readings taken in-line with the hemorrhage site are shown. Average hemorrhage rates measured in the synthetic (**C**) arm and (**D**) leg. Three different bleed rates were recorded using a 20G, 17G, or 14G needle size to adjust the fluidic resistance at the hemorrhage site. Results reported as average values with error bars denoting standard deviation (*n* = 24 for each). Significant differences between hemorrhage rates were determined by one-way ANOVA, post-hoc Tukey’s test (****, *p* < 0.0001).

**Figure 4 sensors-22-01122-f004:**
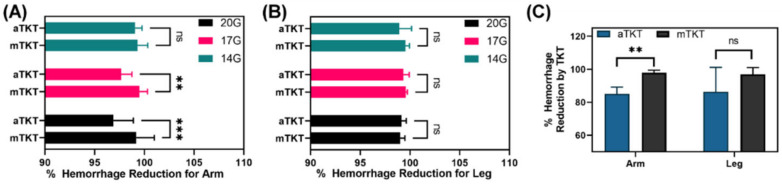
Performance of the aTKT to stop hemorrhage compared to mTKT. Percent hemorrhage reduction 30 s after TKT application for the aTKT and mTKT using the synthetic (**A**) arm or (**B**) leg setup. Average results are shown for 80, 120, 160, and 200 bpm for each of three hemorrhage rates as shown (*n* = 12). Values on the *x*-axis begin at 90% instead of 0% to better highlight differences. Significant differences between hemorrhage rates were determined by two-way ANOVA, post-hoc Sidak’s. (**C**) Percentage hemorrhage reduction five minutes after TKT application for a 14G hemorrhage at 160 bpm (*n* = 3). Significant differences between hemorrhage rates were determined by unpaired *t*-test (**, *p* < 0.01; ***, *p* < 0.005; ns = not significant). Error bars denote standard deviation.

**Figure 5 sensors-22-01122-f005:**
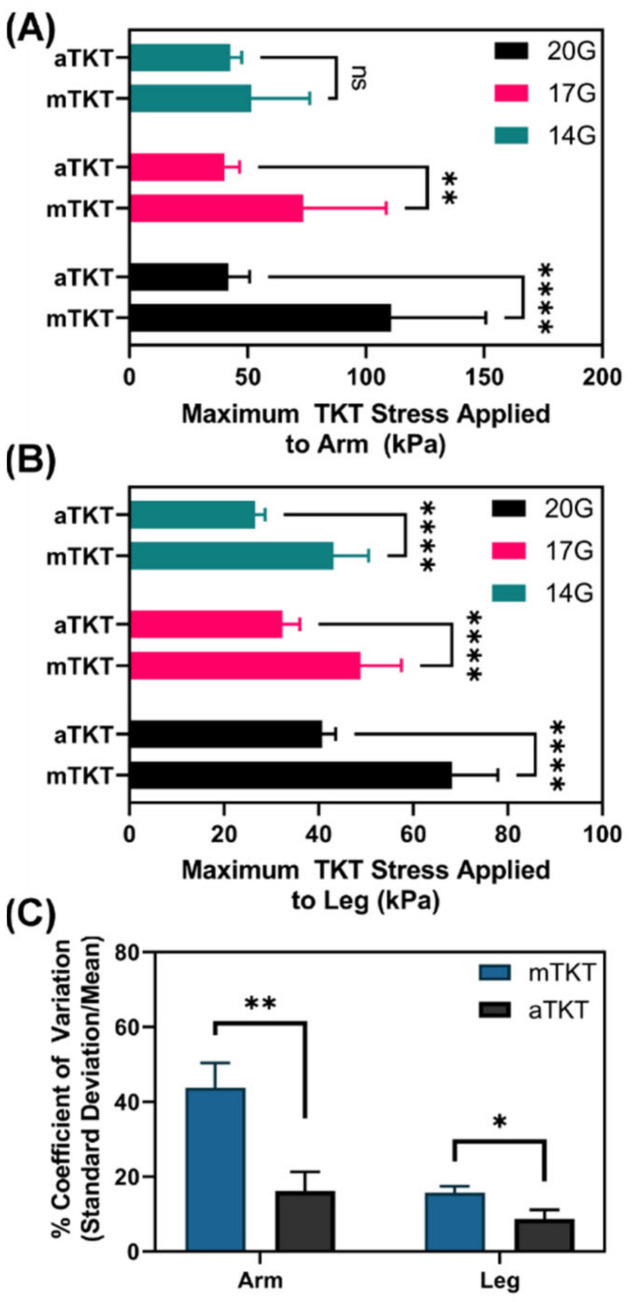
Comparison of TKT application stress for aTKT and mTKT. Maximum application stress (kPa), as determined by force sensor placed under the TKT, for both the aTKT and mTKT using the synthetic (**A**) arm or (**B**) leg test setup. Average results are shown for three hemorrhage rates (*n* = 12). Significant differences between application stresses were determined by two-way ANOVA, post-hoc Sidak’s test. (**C**) Coefficient of variations for the arm and leg TKT stress results for the aTKT and mTKT. Coefficient of variation was calculated as the standard deviation relative to the average for each data set (*n* = 3). Significant differences between application stresses were determined by unpaired *t*-test (*, *p* < 0.05; **, *p* < 0.01; ****, *p* < 0.0001; ns = not significant). Error bars denote standard deviation.

**Figure 6 sensors-22-01122-f006:**
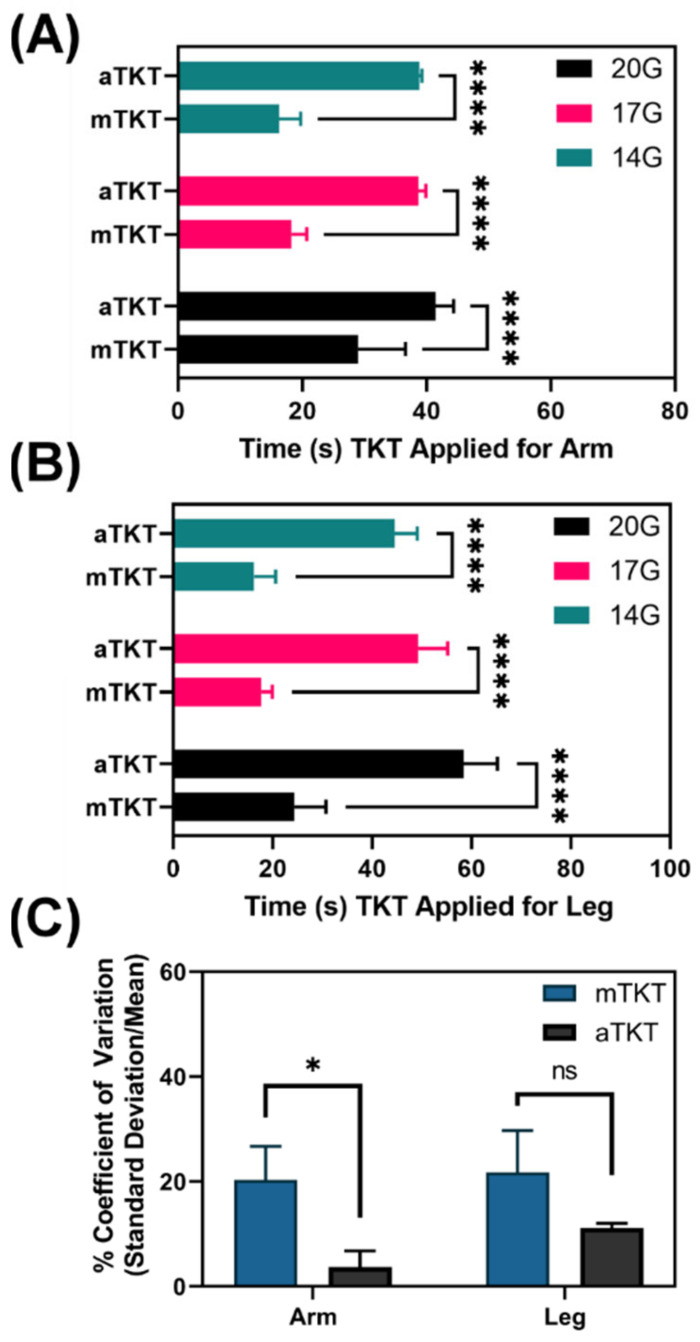
Comparison of TKT application time for aTKT and mTKT. Time for TKT application for both the aTKT and mTKT using the synthetic (**A**) arm and (**B**) leg test setup. aTKT application time was determined by closed loop algorithm while mTKT application was determined by qualitative, visual inspection of hemorrhage stop. Average results are shown for three hemorrhage rates (*n* = 12). Significant differences between application times were determined by two-way ANOVA, post-hoc Sidak’s test. (**C**) Coefficients of variation for the arm and leg TKT application time for the aTKT and mTKT. Coefficient of variation was calculated as the standard deviation relative to the average for each data set (*n* = 3). Significant differences between application times were determined by unpaired *t*-test (*, *p* < 0.05; ****, *p* < 0.0001; ns = not significant). Error bars denote standard deviation.

**Figure 7 sensors-22-01122-f007:**
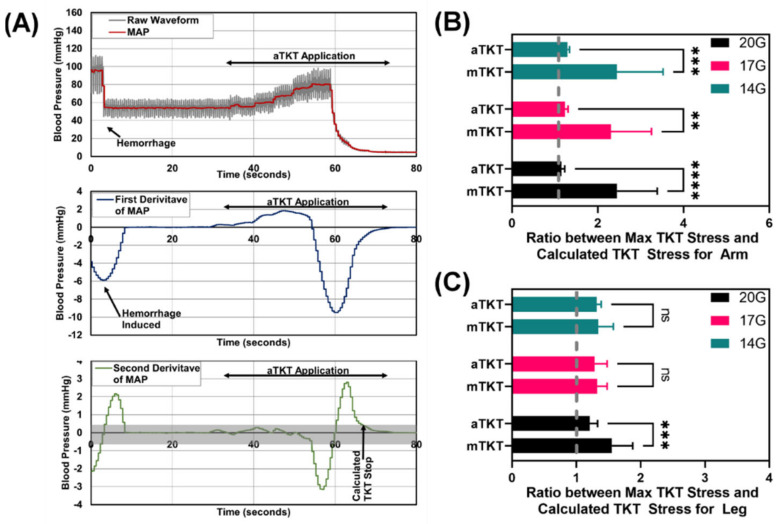
Analysis of Tourniquet overtightening. (**A**) Representative MAP and its first and second derivatives vs. time during aTKT application. (**B**) Ratio between the required TKT application stress at the theoretical bleed stop and maximum TKT application stress for the arm test platform for three hemorrhage rates (*n* = 12). (**C**) Ratio between the required TKT application stress at the theoretical bleed stop and maximum TKT application stress for the leg test platform for three hemorrhage rates (*n* = 12). Significant differences between overtightening ratios were determined by two-way ANOVA, post-hoc Sidak’s test (**, *p* < 0.01; ***, *p* < 0.005; ****, *p* < 0.0001; ns = not significant). Error bars denote standard deviation.

## Data Availability

The datasets generated during and/or analyzed during the current study are available from the corresponding author upon reasonable request.

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
