# Peer review of "Development and Characterization of a Self-Tightening Tourniquet System"

_sensors, 2022, doi:10.3390/s22031122_

Round 1

Reviewer 1 Report

Review of paper:

Development and characterization of a self-tightening tourniquet system

Authors:

Saul J. Vega, Sofia I. Hernandez-Torres, David Berard, Emily N. Boice, Eric J. Snider

Comments:

The research topic presented in the paper is very interesting from the point of view of practical application. The problem of hemorrhage control in both military medicine and civilian medicine has been very important since attempts were made to treat patients with extensive injuries and hemorrhages from the upper and lower extremities. The research for new solutions is therefore the right direction for utilitarian needs. Nevertheless, the paper needs some corrections that will raise its level of content and make the presented content more accessible for Sensors Journal Readers.

List of critical comments below:

  1. In the Introduction section I propose to treat the literature review of the research topic more broadly. In addition, I would ask you to specify there also the general structure of the division of content in the paper, so that the Reader is familiar with the structure of the article from the beginning of reading.
  2. A description of the test platform should be included in the Materials and Method section, which should define the test object, test rig and methodology. In Materials and Method section there should be the general scheme of the developed tourniquet solution with the definition of its component subsystems. This section should be followed by the Results section.
  3. I suggest improving the readability of the posted diagrams. Figure descriptions should be concise. Extensive descriptions there should be in the main text of the paper, not in the figure legend.
  4. The paper does not contain a unified Conclusion section. I propose to add and refer in it to future patient outcomes relative to the results obtained.
  5. In many places in the text the authors use vague terms, stating for example ...was significantly reduced..., significantly less stress…, I therefore propose to make the necessary corrections to many parts of the paper.
  6. The paper should be described in more detail in terms of the measurement results obtained and how they were statistically processed before discussing the results and making charts.
  7. Due to the nature of Sensors Journal, there should be more emphasis on the aspect of sensors used in the research; how they are selected, calibrated and applied for research purposes.
  8. The described algorithms controlling an automatic device could be included in the form of diagrams - it would make them easier to understand.

Reviewer 2 Report

See attached file.

Round 2

Reviewer 2 Report

I thank the authors for their thorough reply to my comments on their manuscript. All questions have been answered and the manuscript has improved a lot. I recommend the manuscript in its present form for publication.

This manuscript is a resubmission of an earlier submission. The following is a list of the peer review reports and author responses from that submission.